Transcriptome-wide identification of WRKY transcription factors and their expression profiles under different stress in Cynanchum thesioides

Chang Xiaoyao
Yang Zhongren
Zhang Xiaoyan
Zhang Fenglan zhangfenglan041105@163.com
Huang Xiumei huangxm0404@126.com
Han Xu
College of Horticulture and Plant Protection, Inner Mongolia Agricultural University , Huhehaote , Inner Mongolia , China
Uversky Vladimir
Electronic publication date: 2022 Dec 2
Publication date: 2022
Volume: 10
Electronic Location ID: e14436
Received 2022 Aug 29; Accepted 2022 Oct 31
Copyright: ©2022 Chang et al.
Copyright year: 2022
Copyright holder: Chang et al.
License: This is an open access article distributed under the terms of the Creative Commons Attribution License, which permits unrestricted use, distribution, reproduction and adaptation in any medium and for any purpose provided that it is properly attributed. For attribution, the original author(s), title, publication source (PeerJ) and either DOI or URL of the article must be cited.
License URL: https://creativecommons.org/licenses/by/4.0/

Keywords: WRKY gene family, Cynanchum thesioides, Plant hormones, Abiotic stresss, Gene expression

Funding: The Inner Mongolia Natural Science Foundation 2020MS03085 The Inner Mongolia Autonomous Region Science and Technology Plan 2021GG0084 The Inner Mongolia Autonomous Region Scientific and Technological Achievements Transformation Special Fund Project 2021CG0023 The Major Science and Technology Project of Inner Mongolia Autonomous Region 2021ZD0001 The Major Science and Technology Projects 2019ZD016 The Inner Mongolia Autonomous Region Applied technology Research and development Project 2019GC237 This work was financially supported by the Inner Mongolia Natural Science Foundation (No. 2020MS03085), the Inner Mongolia Autonomous Region Science and Technology Plan (No. 2021GG0084), the Inner Mongolia Autonomous Region Scientific and Technological Achievements Transformation Special Fund Project (No. 2021CG0023), the Major Science and Technology Project of Inner Mongolia Autonomous Region (No. 2021ZD0001), the Major Science and Technology Projects (No. 2019ZD016) and the Inner Mongolia Autonomous Region Applied Technology Research and Development Project (No. 2019GC237). The funders had no role in study design, data collection and analysis, decision to publish, or preparation of the manuscript.

==============================
Cynanchum thesioides (Freyn) K. Schum. is an important economic and medicinal plant widely distributed in northern China. WRKY transcription factors (TFs) play important roles in plant growth, development and regulating responses. However, there is no report on the WRKY genes in Cynanchum thesioides. A total of 19 WRKY transcriptome sequences with complete ORFs were identified as WRKY transcriptome sequences by searching for WRKYs in RNA sequencing data. Then, the WRKY genes were classified by phylogenetic and conserved motif analysis of the WRKY family in Cynanchum thesioides and Arabidopsis thaliana. qRT–PCR was used to determine the expression patterns of 19 CtWRKY genes in different tissues and seedlings of Cynanchum thesioides under plant hormone (ABA and ETH) and abiotic stresses (cold and salt). The results showed that 19 CtWRKY genes could be divided into groups I-III according to their structure and phylogenetic characteristics, and group II could be divided into five subgroups. The prediction of CtWRKY gene protein interactions indicates that CtWRKY is involved in many biological processes. In addition, the CtWRKY gene was differentially expressed in different tissues and positively responded to abiotic stress and phytohormone treatment, among which CtWRKY9, CtWRKY18, and CtWRKY19 were significantly induced under various stresses. This study is the first to identify the WRKY gene family in Cynanchum thesioides, and the systematic analysis lays a foundation for further identification of the function of WRKY genes in Cynanchum thesioides.

Introduction

WRKY transcription factors are widely present in plants and are one of the largest and most unique families of transcription factors in plants (Xie et al., 2018). WRKY transcription factors share a highly conserved WRKYGQK core motif at their N-terminal end, in which the amino acids W, K and Y are highly conserved (Eulgem et al., 2000), although mutants with substituted amino acid residues have also been reported (Li et al., 2016). Moreover, there are two types of zinc finger structures at the C-terminus of the WRKY structural domain, C2HC (C-X7-C-X23-HX-C) and C2H2 (C-X4-5-C-X22-23H-X-H) (Li et al., 2016), and the WRKY family is divided into three subfamilies based on the number of WRKY structural domains and zinc finger structure characteristics: subfamily class I has two WRKY structural domains, and this class of proteins has C2H2-type zinc finger structures (Eulgem et al., 2000). The zinc finger structure in the class II subfamily is also a C2H2 type but has only one WRKY domain, while the class III subfamily contains one WRKY structural domain and a C2HC-type zinc finger structure (Song et al., 2014).

Plants survive in complex and variable environments and are often subjected to various biotic and abiotic stresses, and WRKY transcription factors have important roles in plant growth and development and in coping with adverse stresses (Chen et al., 2020a; Chen et al., 2020b). For example, WRKY transcription factors affect lignin synthesis (Miyamoto et al., 2020) and fruit ripening (Gan et al., 2021) during plant growth and development; additionally, WRKY TFs are involved in plant responses to drought (Lim et al., 2022), salt (Fang et al., 2021), and low temperature (Zhang et al., 2019a) and a variety of other stress responses. It has been shown that WRKY transcription factors initiate a response in plants mainly by interacting with the cis-acting element W-box on the target gene promoter, which acts as a transcriptional activator or repressor of downstream gene expression (Bakshi & Oelmüller, 2014). For example, VQ20 in Arabidopsis interacts with WRKY2 and WRKY34 to coregulate pollen development (Lei et al., 2017). SIB1 and SIB2 interact with the AtWRKY3 protein to play an activating role in Arabidopsis defence against pathogenic bacteria (Lai et al., 2011).

Cynanchum thesioides (Freyn) K. Schum. belongs to the Cynanchum genus of the Asclepiadaceae family. It is an upright, xerophytic shrub; the underground stem is simple and transverse; and the stem is branched from the base (Qian & Chen, 1959). In China, Cynanchum thesioides is distributed in Inner Mongolia, Northeast China, and Xinjiang and grows in arid sands, wastelands, and field margins (Zhang et al., 2019b). It can be used as a medicine, food, forage resource and for wasteland and barren land management (Zhang et al., 2019b). However, as the environment continues to deteriorate, Cynanchum thesioides face different biotic and abiotic stresses. Their populations are gradually decreasing, especially in Inner Mongolia, China, due to the cold weather and land salinization. Therefore, the screening of anther development-related genes and stress-related genes is of great significance for creating male sterile germplasm resources and the artificial breeding of fine varieties of Cynanchum thesioides. WRKY genes have important roles in plant biotic and abiotic stresses (Chen et al., 2020b); therefore, identification of their members is necessary to better understand the function of WRKY genes in Cynanchum thesioides. In this study, we screened WRKY genes using transcriptome data and analysed the phylogeny, domain structures, and expression patterns of the screened WRKY genes under different abiotic stresses and phytohormone treatments. This study provides basic information for the identification and classification of CtWRKY genes and for further studies on the functional regulation mechanism of CtWRKY genes under various stresses.

Materials & Methods

Identification of WRKY genes in Cynanchum thesioides

Due to the lack of genomic information, we used different transcriptome data to identify the WRKY gene family to ensure sequence accuracy. The gene sequences of Cynanchum thesioides were obtained from drought stress transcriptome data from previous studies (Zhang et al., 2019b) and unpublished transcriptome data of different anther development from the laboratory. The Hidden Markov Model (HMM) file (PF03106) for the WRKY domain was downloaded from the Pfam database (https://www.ebi.ac.uk/interpro) (Punta et al., 2004). In order to screen out WRKY candidate genes, BLASTp detection was performed on the Cynanchum thesioides transcriptome data using HMM with the e-value set to 0.01. Validation of candidate WRKY genes was performed using an online program (http://pfam.xfam.org/search) containing a conserved domain database search (https://www.ncbi.nlm.nih.gov/Structure/cdd/wrpsb.cgi) and an online tool (http://smart.embl-heidelberg.de/) to delete genes without WRKY domains. Sequences with 97% or more similarity between different transcriptome databases were manually removed. The search results were predicted by searching open reading frames (ORFs) (https://www.ncbi.nlm.nih.gov/gorf/gorf.html) and subsequent experiments were performed on genes with complete ORFs.

Sequence analyses of Cynanchum thesioides

The physical properties of WRKY members, such as length, molecular weight (MW), and isoelectric point (pI), were predicted using the online ExPasy program (http://www.expasy.org/tools/) (Wilkins et al., 1999).

We analysed the conserved motifs using the MEME website (https://meme-suite.org/meme/) and annotated with InterPro Scan (http://www.ebi.ac.uk/interpro/). The conserved motif of CtWRKY was detected by the online tool MEME (https://meme-suite.org/meme/) with parameters set to 0 or 1 occurrence per sequence, up to 10 subjects, and the range of the motif length was set to 5-50 aa (Bailey et al., 2015).

Phylogenetic analysis of Cynanchum thesioides

The amino acid sequence of CtWRKY with a complete ORF was compared with the WRKY gene of Arabidopsis thaliana by multiple sequence alignment using the Clustal Omega tool. Based on the comparison results, we constructed phylogenetic trees using the maximum likelihood method using MEGAX with 1,000 bootstrap replications of maximum likelihood (Tamura et al., 2011). The online ITOL (http://itol.embl.de/help.cgi) tool was used to embellish the phylogenetic tree.

Interaction network analysis of CtWRKY proteins

An interaction network was constructed for all CtWRKY. The homologs of each CtWRKY in Arabidopsis were identified using BLAST software (Chen et al., 2020a). After that, the Arabidopsis homologs corresponding to each CtWRKY gene were submitted to the STRING database (https://string-db.org/cgi/input.pl) (Szklarczyk et al., 2017) with default parameters to construct the interaction network.

Plant materials and treatments

The seeds of Cynanchum thesioides were collected at the Inner Mongolia Agricultural University, Huhot, Inner Mongolia Province, Northwest China (111°69′E, 40° 80′N), and seedlings were raised according to a previous method (Zhang et al., 2019b) and used for each assay in this paper.

Seedlings were treated with abiotic stress and hormones, including 4 °C (cold), 150 mM NaCl (salt), 0.1 mM abscisic acid (ABA), and 500 µM ethephon (ETH) at the ten-leaf stage. Leaves were collected at 0 h, 12 h, and 24 h after treatment and used to analyse the expression patterns of the different treatments. Then, Cynanchum thesioides roots, stems, leaves, flowers, fruits, and anthers at different developmental stages (according to the different lengths of the flower buds, it is named the T1-T4 stage) were collected to analyse the expression patterns of different tissues (Fig. 1). All samples were immediately frozen in liquid nitrogen and stored at −80 °C for subsequent extraction of total RNA. Three replicates were used for all samples.

RNA extraction and quantitative real-time PCR (qRT −PCR)

Total RNA was extracted from Cynanchum thesioides using an RNA kit (TIANGEN BIOTECH, Beijing, China) according to the manufacturer’s instructions. First-strand cDNA was synthesized using TransScript One-Step gDNA Removal and cDNA Synthesis SuperMix (Transgen Biotech, Beijing, China). Real-time fluorescent quantitative PCR (qRT −PCR) was performed using TB Green Premix Ex Taq II (RR420Q TaKaRa Biotechnology, Beijing, China) on an FTC-3000P (Funglyn Biotech, Toronto, Canada) system for real-time fluorescence quantitative PCR (qRT −PCR), and the primers are shown in Table S1. The reaction procedure was completed under the following program: 60 s of predenaturation at 95 °C, 40 cycles of 15 s at 95 °C and 15 s at 60 °C, and a final hold at 4 °C. All samples were assayed using three replicates. Relative expression levels were calculated using the 2−ΔΔCT method (Livak & Schmittgen, 2001). By screening four reference genes of Cynanchum auriculatum, we found that ACT7 was the most stable under different abiotic stresses and hormone treatments and that GAPDH was the most stable in different tissues, so we selected ACT7 and GAPDH as internal control (Zhu et al., 2022).

Figure 1 The materials used to detect the CtWRKY gene expression.

The different tissues used for total RNA extraction, including stem (A), leaf (B), flower (C), fruit (D), root (E) and different developmental stages of the anther (F). The blue line indicates the position at which stem and leaf tissues were collected.

Results

Identification and characteristics of the WRKY gene family

WRKY structural domains were analysed by an HMMER search for candidate genes using SMART and NCBI CDD tools, confirming that these genes contain single or double WRKY structural domains and indicating that they are indeed members of the WRKY gene family. 96 WRKY genes were predicted by searching open reading frames (ORFs) in C. thesioides. Finally, 19 WRKY genes with complete ORFs were obtained. Since there is no standard nomenclature for WRKYs, the 19 WRKYs were designated CtWRKY1 to CtWRKY19 based on their gene IDs. A similar nomenclature is found in Pinus massoniana Lamb (Yao et al., 2020). All of these CtWKRY genes more conserved WRKYGQK structural domain and a C2H2 or C2HC-type zinc finger motif. CtWRKY ORFs range in length from 172aa (CtWRKY1) to 569aa (CtWRKY14), in molecular weight from 18.8 kDa (CtWRKY1) to 63.0 kDa (CtWRKY14), and in predicted isoelectric point from 4.81 (CtWRKY5) to 9.71 (CtWRKY10). Subcellular localization predictions showed that all CtWRKY genes were localized in the nucleus (Table 1).

Phylogenetic analysis and conserved motif detection of CtWRKY proteins

Based on previous studies, 69 AtWRKY proteins from different subgroups were randomly selected as representatives of Arabidopsis for comparison with Cynanchum thesioides (Eulgem et al., 2000; Yao et al., 2020). The most distinctive feature of WRKY proteins is the WRKY structural domain containing 60 amino acids, including the highly conserved feature “WRKYGQK” and a C2H2- or C2HC-type zinc finger motif (Li et al., 2016). The results showed that the structural domain sequences of CtWRKY all contained the conserved heptapeptide “WRKYGQK” (Fig. 2). Phylogenetic trees were constructed based on the WRKY protein sequences of Arabidopsis and Cynanchum thesioides. The results showed that the 19 CtWRKYs were classified into three categories (I-III) according to the taxonomic criteria of Arabidopsis, with seven CtWRKYs in group I, eight CtWRKYs in group II, and four CtWRKYs in group III. Group II had a further division into five categories, with one CtWRKY in IIa, two each in IIb and IId, none in IIc, and three in IIe (Fig. 3).

Table 1 The statistic information of CtWRKYs in Cynanchum thesioides.

The physical properties of WRKY members, such as length, molecular weight (MW), isoelectric point (pI), WRKY domain, zinc-finger type, and prediction of subcellular localization.

Gene ID	Gene	ORF (aa)	PI	MW (kDa)	WRKY Domain	Zinc- finger type	Subcellular localization	Clusters	
BMK_Unigene_015293	CtWRKY1	172	8.87	18.8	WRKYGQK	C2H2	Nucleus	GroupII-b	
BMK_Unigene_016135	CtWRKY2	432	6.68	47.3	WRKYGQK/WRKYGQK	C2H2	Nucleus	GroupI	
BMK_Unigene_033686	CtWRKY3	503	6.33	54.9	WRKYGQK/WRKYGQK	C2H2	Nucleus	GroupI	
BMK_Unigene_034217	CtWRKY4	364	9.41	39.6	WRKYGQK	C2H2	Nucleus	GroupII-d	
BMK_Unigene_093634	CtWRKY5	392	4.81	42.7	WRKYGQK	C2H2	Nucleus	GroupII-e	
BMK_Unigene_139370	CtWRKY6	340	6.10	37.8	WRKYGQK	C2HC	Nucleus	GroupIII	
BMK_Unigene_140777	CtWRKY7	533	8.83	58.4	WRKYGQK/WRKYGQK	C2H2	Nucleus	GroupI	
BMK_Unigene_141975	CtWRKY8	370	9.67	40.6	WRKYGQK	C2H2	Nucleus	GroupII-d	
BMK_Unigene_142518	CtWRKY9	326	6.07	36.0	WRKYGQK	C2H2	Nucleus	GroupII-a	
BMK_Unigene_144375	CtWRKY10	308	9.71	33.2	WRKYGQK	C2H2	Nucleus	GroupII-b	
Cluster-6587.14962.p1	CtWRKY11	384	5.81	41.6	WRKYGQK	C2H2	Nucleus	GroupII-e	
Cluster-6587.21502.p1	CtWRKY12	566	5.05	61.9	WRKYGQK/WRKYGQK	C2H2	Nucleus	GroupI	
Cluster-6587.26338.p1	CtWRKY13	470	6.33	51.6	WRKYGQK/WRKYGQK	C2H2	Nucleus	GroupI	
Cluster-6587.27607.p1	CtWRKY14	569	6.34	63.0	WRKYGQK/WRKYGQK	C2H2	Nucleus	GroupI	
Cluster-6587.30041.p1	CtWRKY15	440	8.79	48.5	WRKYGQK/WRKYGQK	C2H2	Nucleus	GroupI	
Cluster-6587.47842.p1	CtWRKY16	359	5.58	40.3	WRKYGQK	C2HC	Nucleus	GroupIII	
Cluster-6587.48940.p1	CtWRKY17	261	5.48	28.5	WRKYGQK	C2H2	Nucleus	GroupII-e	
Cluster-6587.66182.p1	CtWRKY18	343	5.30	38.9	WRKYGQK	C2HC	Nucleus	GroupIII	
Cluster-6587.79872.p1	CtWRKY19	233	5.21	26.3	WRKYGQK	C2HC	Nucleus	GroupIII	

Figure 2 Multiple sequence alignment of CtWRKY domains.

The sequences of WRKY variants were aligned. The conserved WRKYGQK and zinc-finger residues were marked in gray. Above the sequence is the sequence logos of the conserved motif.

Figure 3 Phylogenetic tree of WRKY family in Cynanchum thesioides.

Different color branches and strips are used to distinguish different subgroups. In addition, the asterisks indicate Cynanchum thesioides. The maximum likelihood method was used to analyze the evolutionary trees of 19 CtWRKYs and 69 AtWRKYs.

To study the conserved motifs of WRKY proteins in Cynanchum thesioides, we analysed the conserved motifs using the MEME website and annotated with InterPro Scan. A total of 10 predicted conserved motifs, called motifs 1–10, were identified with amino acid lengths ranging from 9–50, and the number of motifs in each protein sequence ranged from two to ten. The first four motifs were annotated as DNA-binding domain (Fig. 4). Among these motifs, motif 1 is an important structure of the WRKY structural domain present in all CtWRKY proteins. In addition to motif 1, motif 2 is also widely available in most members. The higher number of motifs in group I compared with other subgroups and the presence of motif 4 only in group I suggest that these genes may have multiple functions. Within a subpopulation, CtWRKY proteins have similar motifs and are relatively conserved. In addition, Although CtWRKY9 belongs to group II a, it shares a similar conserved motif with groups II b and II d.

Figure 4 Conserved motifs analysis of the CtWRKY proteins.

We analysed the conserved motifs using the MEME website and annotated with InterPro Scan. The different colored boxes represent different motifs and their position in each CtWRKY sequence. Each motif is indicated by a colored box in the legend at the bottom.

Interaction network analysis of CtWRKY proteins

To better understand the potential interactions of CtWRKY proteins, interaction networks CtWRKY proteins were constructed using STRING software. These WRKY-interacting proteins mainly include VQ proteins (MKS1) involved in regulating plant defense responses, MPK3 and MPK4 proteins involved in plant responses to pathogens and stress, IKU1 proteins involved in regulating endosperm growth and seed size, and other proteins (Petersen et al., 2010; Berriri et al., 2012; Yan et al., 2021; Wang et al., 2010). Figure 5 shows the predicted interaction network between the 11 CtWRKY proteins VQ and MPK proteins. Among them, AtWRKY33 (CtWRKY2) had a strong interacts with 10 VQ proteins and 3 MPK proteins. In addition, AtWRKY53 (CtWRKY16), AtWRKY40 (CtWRKY9), and AtWRKY70 (CtWRKY6) interact with multiple VQ proteins to form a complex regulatory network. Previous studies suggest that there are interactions between WRKY proteins (Chi et al., 2013). Through interaction networks, we found that CtWRKYs are involved in biotic stress response and defence (AtWRKY33/CtWRKY2, AtWRKY40/CtWRKY9, AtWRKY53/CtWRKY16) (Zheng et al., 2006; Lozano-Durán et al., 2013; Wang et al., 2020), abiotic stress response (ZAP1/CtWRKY14) (Qiao, Li & Zhang, 2016), SA and JA signalling (AtWRKY22/CtWRKY11) (Kloth et al., 2016), salicylate signalling (AtWRKY70/CtWRKY6) (Ulker, Shahid Mukhtar & Somssich, 2007), pollen development (AtWRKY2/CtWRKY12) (Lei et al., 2017), and other biological processes. These results suggest that CtWRKYs may play important roles in various biological processes, which may provide important clues for understanding and validating the role of CtWRKYs in the response to various stresses.

Figure 5 Interaction networks of the CtWRKY proteins.

Protein–protein interaction networks of CtWRKYs as predicted by STRING search tool.

Expression pattern of the CtWRKY genes in different tissues

To investigate the possible role of the CtWRKY gene in the growth and development of Cynanchum thesioides, we performed qRT–PCR expression analysis of five tissues of Cynanchum thesioides, root, stem, leaf, flower, and fruit, as well as anthers at different developmental stages (Fig. 6). The results showed that most CtWRKY genes were highly expressed in at least one tissue and only at low levels in other tissues; a result that may be related to the interaction of WRKY transcription factors with other genes or proteins during plant growth and development. There were 11, five, and eight WRKY genes highly expressed in the roots, stems, and flowers of Cynanchum thesioides. Respectively, CtWRKY5 and CtWRKY12 are the most highly expressed in roots and CtWRKY2, CtWRKY7, and CtWRKY18 are the most highly expressed in flowers. Only four genes were highly expressed in fruits, while only two were expressed in leaves. These 19 CtWRKY genes were expressed in different tissues, indicating that CtWRKY genes have a tissue-specific expression profile.

Figure 6 Expression profile of CtWRKY genes in various tissues.

Expression data were normalized based on the mean expression value of each gene in all tissues/organs. Different tissues are displayed vertically below each column. Gene names are displayed to the right of each row.

19 CtWRKY genes were analysed by qRT–PCR to determine their expression at different developmental stages of the anther (Fig. 7). The results showed that five genes (CtWRKY2, CtWRKY8, CtWRKY9, CtWRKY12, and CtWRKY17) were downregulated in expression more than five-fold at T2, and six genes (CtWRKY6, CtWRKY7, CtWRKY10, CtWRKY15, CtWRKY18, and CtWRKY19) were expressed at T3. The expression of two genes (CtWRKY1 and CtWRKY14) were highly upregulated by six-fold and five-fold at T4 (Fig. 7). The results suggest that the WRKY gene family plays an important role in the development of Cynanchum thesioides anthers.

Figure 7 CtWRKY genes were analysed by PCR to determine their expression at different developmental stages of the anther.

The relative expression levels of all CtWRKYs are calculated by 2−ΔΔCt method, and T1 stage is used as the standard control. The gene name is on the top left of each column graph. Asterisks indicate statistically significant differences between the stressed samples and counterpart controls (* p < 0.05, ** p < 0.01).

Expression pattern of the CtWRKY genes under ETH and ABA treatments

It has been shown that ETH and ABA play important roles in regulating plant development, stress responses, and various physiological processes (Lin, Zhong & Grierson, 2009; Ku et al., 2018). Therefore, to investigate the role of the CtWRKY genes in response to ETH and ABA, we analysed the expression pattern of the CtWRKY genes under ETH and ABA treatment by qRT–PCR. The results showed that three genes (CtWRKY7, CtWRKY18, and CtWRKY19) were upregulated at 12 h and 24 h under ETH treatment, with expression up to 10-fold compared with 0 h. Two genes (CtWRKY2 and CtWRKY9) showed a trend of upregulation followed by downregulation, peaking at 12 h, which was three times higher than that at 0 h. In contrast, four genes (CtWRKY1, CtWRKY8, CtWRKY16, and CtWRKY17) showed a significant decreasing trend after ETH treatment (Fig. 8A).

Figure 8 Expression profile of the CtWRKYs in response to plant hormones and abiotic stresss.

(A) ETH, (B) ABA, (C) cold, (D) salt. The color scale represents relative expression levels based on the values of log2 (2−ΔΔCt). Asterisks represent significant differences between each time point and 0 h (* p < 0.05, ** p < 0.01).

In addition, we found that CtWRKY18 showed the same expression pattern under ABA treatment as under ETH treatment, and other CtWRKY genes showed significant dynamic changes when subjected to ABA treatment. Among them, four genes (CtWRKY9, CtWRKY11, CtWRKY14, and CtWRKY15) showed significantly upregulated expression, while two genes (CtWRKY6 and CtWRKY8) showed significantly downregulated expression. In addition, CtWRKY1 and CtWRKY19 were significantly upregulated at 12 h, whereas CtWRKY13 and CtWRKY16 expression reached the highest level at 12 h and then decreased, but it remained significantly higher than that at 0 h (Fig. 8B).

Expression pattern of the CtWRKY genes under salt and cold stress

Abiotic stresses severely affect plant growth and geographic distribution. To further investigate the potential role of the CtWRKY genes under abiotic stress, we analysed the expression pattern of the CtWRKY genes under low temperature and salt stress by qRT–PCR. The results showed that all CtWRKY genes responded positively under low-temperature stress treatment. Among them, five genes (CtWRKY2, CtWRKY5, CtWRKY6, CtWRKY7, and CtWRKY10) showed a gradual downregulation under low-temperature stress induction. CtWRKY8 and CtWRKY13 were significantly upregulated at 24 h and were four-fold higher than the control (0 h). In contrast, CtWRKY18 and CtWRKY19 were significantly upregulated at all time points and reached the highest expression at 24 h, which was 30-fold higher than the control (0 h) (Fig. 8C). In addition, CtWRKY18 maintained the same expression pattern under salt stress treatment. On the other hand, CtWRKY3, CtWRKY9, and CtWRKY19 were induced by salt stress and reached the highest expression level at 12 h of treatment, which was 20-fold higher than the control (0 h). Four genes (CtWRKY1, CtWRKY8, CtWRKY16, and CtWRKY17) were significantly downregulated at various time points of salt stress treatment (Fig. 8D).

Discussion

The WRKY gene family is widespread in plants, and its members play important roles in a variety of plant biological processes (Ulker & Somssich, 2004). To date, the WRKY gene family has been identified in a variety of plants, including Arabidopsis (Dong, Chen & Chen, 2003), rice (Ross, Liu & She, 2007), tomato (Huang et al., 2012), and eggplant (Yang et al., 2020). However, the WRKY gene family has not been reported in Cynanchum thesioides, which has important economic and medicinal uses. Based on the available transcriptomic data (Zhang et al., 2019b), research on this topic is very limited due to the lack of genomic information and previous in-depth studies. Therefore, we used the transcriptome to identify the WRKY gene family here. A total of 96 WRKY genes were found in the transcriptome, but only 19 of them had complete ORFs. we performed a comprehensive identification of the WRKY gene in Cynanchum thesioides and further investigated its sequence characteristics, protein structure, subcellular localization, and expression pattern. This study provides the basis for subsequent functional analysis of CtWRKY genes.

We identified 19 CtWRKY genes with complete ORFs from Cynanchum thesioides and found that their ORF lengths ranged from 172 to 569 bp (Table 1), which is consistent with previous studies (Dong, Chen & Chen, 2003). In addition, most CtWRKY genes produce acidic proteins with isoelectric point less than 7, a phenomenon commonly found in WRKY proteins from other plants (Huang et al., 2012; Yang et al., 2020). Based on the number of conserved WRKY structural domains and the type of zinc finger structures, WRKY gene family members can be divided into groups I, II, and III. Evolutionarily, WRKY members in group I are the ancestors of members in the other groups, and the gene functions of group I members are more conserved than those of groups II and III (Cheng et al., 2019; Villano et al., 2020). In Cynanchum thesioides, the lowest number of CtWRKY members was found for group III, and the highest number was found for group II, suggesting that group II CtWRKY genes are more evolutionarily active. This is in contrast to what was reported in rice (Wu et al., 2005) and may be related to the different evolutionary processes and strategies in different plants. The conserved motif distribution pattern is the main basis for the classification of gene family members. Motif 1 is present in all WRKY family members. This sequence is essential for the WRKY transcription factor to recognize the binding of the W-box element on the promoter of the target gene (Zhang & Wang, 2005). CtWRKY genes in the same subgroup have similar conserved motif distribution patterns, while the conserved motif distribution patterns of CtWRKY genes in different subgroups differ. It is hypothesized that CtWRKY genes perform similar or different biological functions due to the differences and similarities in conserved motif distribution patterns.

Numerous studies have found that WRKY genes are expressed in one or more tissues and play a key role in regulating plant growth and development (Wang et al., 2019a). Analysis of the expression patterns of CtWRKY genes in different tissues revealed that 19 CtWRKY genes were highly expressed in all five tissues of Cynanchum thesioides, which may indicate that these genes play important roles in the development of various tissues of Cynanchum thesioides. The expression of CtWRKY gene family members in different groups varied greatly, suggesting that they may have different functions, and the wide range of expression in group I indicates that it plays a key role in growth and development. Most WRKY gene family members are expressed at higher levels in roots than in other tissues, and these genes may be involved in some regulatory mechanisms in roots (Fig. 6). It has been shown that gene expression during pollen development is divided into early and late stages (Verelst et al., 2007; Honeys et al., 2006). AtWRKY34 was identified as an “early gene” that is transiently phosphorylated by MPK3/MPK6 during the early stages of pollen development and then dephosphorylated and degraded before pollen maturation (Guan et al., 2014). VQ proteins interact with the WRKY transcription factors and play a critical role in plant stress response and growth and development (Zhang et al., 2022). CtWRKY12 and AtWRKY34 clustered together and showed similar expression patterns, and it is speculated that these genes may have similar functions. On the other hand, AtWRKY2 and AtWRKY34 play redundant roles in pollen development; however, VQ20 interacts with AtWRKY34 and AtWRKY2 to regulate the function of WRKY in pollen development (Lei et al., 2017). In the interaction network prediction of CtWRKY proteins, we found that CtWRKY12 is homologous to AtWRKY2 and interacts with VQ20, further suggesting that CtWRKY12 may have some functions during pollen development. In addition, similar expression patterns existed for CtWRKY2, CtWRKY9, and CtWRKY17. Six CtWRKY genes (CtWRKY6, CtWRKY7, CtWRKY10, CtWRKY15, CtWRKY18, and CtWRKY19) were specifically overexpressed at T3 during anther development (Fig. 7). This will provide a theoretical basis for studying the function of the CtWRKY gene. Similar genes are present in cotton, where GhWRKY22 is mainly expressed in late developing anthers/pollen and transgenic Arabidopsis plants exhibit male fertility defects and low pollen viability (Wang et al., 2019b). These late genes may also be used as candidate genes for regulating anther/pollen development.

Many studies have shown that WRKY also plays an important role in regulating plant adaptation to adversity (Chen et al., 2020b). Gene expression patterns can provide important clues to studying the functions of genes, so we examined the expression patterns of CtWRKY genes in leaves of Cynanchum thesioides under different treatments using the qPCR technique. Six genes (CtWRKY3, CtWRKY9, CtWRKY11, CtWRKY13, CtWRKY18, and CtWRKY19) responded to at least two stresses, with CtWRKY18 and CtWRKY19 being significantly elevated in response to ABA, ETH, cold, and salt treatments. Interestingly, the expression of CtWRKY18 peaked at 24 h under all treatments (Fig. 8), suggesting that this gene may have temporal specificity. The AtWRKY30, closely related to WRKY18 and WRKY19, is expressed under high salinity and ABA induction (Zou et al., 2019). At the same time, AtWRKY30 overexpression plants significantly improve the resistance of Arabidopsis to salt stress (Scarpeci et al., 2013). In this study, CtWRKY18 and CtWRKY19 were significantly up-regulated under salt stress and ABA treatment (Figs. 8B, 8D), which was consistent with the stress response function of the homologous gene AtWRKY30. In addition, homologous WRKY genes may have evolved different functions in different plants (Song et al., 2014). In transgenic Arabidopsis thaliana, AtWRKY17 enhances tolerance to salt stress (Ali et al., 2018). However, the same subgroup of CtWRKY8 genes showed down-regulated in salt stress. WRKY40 is a negative regulator that inhibits the expression of ABA-responsive genes. However, high levels of ABA signaling inhibited WRKY40 expression (Shang et al., 2010). The homologous gene CtWRKY9 in Cynanchum thesioides showed the opposite expression pattern; therefore, the function of the gene needs to be verified in additional experiments. In the present study, we found that CtWRKY9, CtWRKY18, and CtWRKY19 were highly expressed under various treatments, which deserves further investigation.

Conclusion

In Cynanchum thesioides, we identified a total of 19 CtWRKY genes with complete ORFs that were classified into three groups based on conserved motifs. The possible functions of CtWRKY genes were revealed by phylogenetic analysis and protein interaction prediction, and candidate genes were screened by gene expression analysis. RT–qPCR further validated the key role of 19 CtWRKYs under abiotic stresses (salt and cold), different hormone stresses (ABA and ETH), and different developmental stages of the anthers in Cynanchum thesioides. This study provides a theoretical basis for the functional study of CtWRKY genes and a potential strategy for breeding Cynanchum thesioides.

Supplemental Information

Supplemental Information 1 Raw Data

Click here for additional data file.

Additional Information and Declarations

Competing Interests

Author Contributions

Data Availability

The authors declare there are no competing interests.

Xiaoyao Chang conceived and designed the experiments, performed the experiments, analyzed the data, prepared figures and/or tables, authored or reviewed drafts of the article, and approved the final draft.

Zhongren Yang conceived and designed the experiments, authored or reviewed drafts of the article, and approved the final draft.

Xiaoyan Zhang conceived and designed the experiments, authored or reviewed drafts of the article, and approved the final draft.

Fenglan Zhang conceived and designed the experiments, analyzed the data, prepared figures and/or tables, authored or reviewed drafts of the article, and approved the final draft.

Xiumei Huang conceived and designed the experiments, analyzed the data, prepared figures and/or tables, authored or reviewed drafts of the article, and approved the final draft.

Xu Han conceived and designed the experiments, authored or reviewed drafts of the article, and approved the final draft.

The following information was supplied regarding data availability:

The raw measurements are available in the Supplementary File.

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
