# Peer review of "Transcriptome-wide identification of WRKY transcription factors and their expression profiles under different stress in Cynanchum thesioides"

_PeerJ, doi:10.7717/peerj.14436_

## Round 0.1 · original submission · Major Revisions

As you can see, the recommendations of the reviewers are ranging from minor revision to reject. Please address all the concerns of the reviewers and amend manuscript accordingly.

·

Basic reporting

The manuscript is written well overall. It is clear and used professional English. Sufficient background significance are given by the authors in the introduction section. Literature references are sufficient. Article is structured with all requirements - abstract, introduction, materials and methods, results, discussion, conclusion, references, figures and tables. The manuscript is self-contained with relevant results to the hypotheses proposed in the introduction.

Experimental design

The research is within in the Aims and Scope of the journal. Research question - characterization of WRKY gene family, its role in stress tolerance is well defined. The authors have stated that how this research work fills the knowledge gap in this subject area. Experiments are designed with high standard - bioinformatic analysis and the wet lab work on stress, hormone treatments and gene expression studies. All the experiments are described in detail mentioning the steps, databases and tools used with proper references. Required replications are maintained. Why cold and salt stress alone were chosen? Any background information about its relevance to the plant as recorded in the literature based on any field studies may be included to support the selection of abiotic stresses. Similarly, materials and methods SA and ETH are mentioned. But in results and discussion JA is missing but ABA is mentioned. This may be checked.

Validity of the findings

All the data are provided and they are statistically sound. Findings are valid. But the total of 19 genes is a small number. There may be many WRKY genes in the genome. Somewhere in the discussion section justification can be included. WRKY genes are transcription factors. Hence some more discussion on the downstream genes that would have been transcribed by WRKYs, their functional significance in observed stress tolerance / hormone response may be included. Conclusions are well stated and linked to original research question.

Additional comments

At some places of the manuscript the plant is mentioned as 'groundnut' (lines 75, 118, 255 etc). Hope this is the common name of the plant in the country and not typo error.

·

Basic reporting

1. BASIC REPORTING
The manuscript titled “Transcriptome-wide identification of WRKY transcription factors and their expression profiles under different stress in Cynanchum thesioides” is mostly a general bioinformatics prediction of WRKY genes from transcriptome, their motif analysis, phylogenetic analysis, and their expression under various stresses. The manuscript does not offer any novel scientific content and analysis. It describes a very loose in-silico analysis without going into the depth of the analyses.

Experimental design

The tools and methods used in this study are not well described. For instance.
The authors used ClustalW for the multiple sequence alignment. A more recent version called Clustal Omega should be used for multiple sequence alignment. It is more reliable.
Moreover, what parameters were used for the construction of phylogenetic tree i.e. complete deletion, partial deletion or pair-wise deletion.
Instead of preparing the phylogenetic tree with Neighbour-Joining method, a Maximum Likelihood method is more reliable.
The authors described the identification of various domains using MEME. The authors did not mention
On what basis the 10 number of motifs were selected?
What size of the motif was selected for the identification of these domains in WRKY proteins? Moreover, did the authors try to annotate the motifs identified through MEME to gain insights in the functions of these motifs?
Line 118: What are the groundnut root?

Validity of the findings

The Results are weakly described, and very weak discussion of the results.
Line: 314: Is this study carried out on groundnut?

Additional comments

The English language should be improved to ensure that an international audience can clearly understand your text.
The nomenclature for writing the name of genes and proteins in the text is not uniform. Sometimes the genes are written in italics, sometimes in normal.
The authors should also validate the functions of a few of these genes through the development of mutant or over-expression lines.
Such weaknesses in the Manuscript make it a very general study and don’t stand with the objective of novelty and/or biological significance.

Reviewer 3 ·

Basic reporting

The manuscript highlights characterizing of Wrky transcriptomic sequences and carrying out bioinformatic and experimental analysis to subgroup them as well as obtain more insights about the family. Although the study is interesting, major work needs to be done on it to make it through in publishable format. English needs to be edited throughout the manuscript, there are no major grammatical errors/typos at all, but the way content needs to be framed in a manner so that it is easy to comprehend what the study is about and what has been done.
Here are some points which need attention :

-For the Supplementary Tables(starting with the name Raw data), please make the legends self-explanatory. It is not clear what data is been shown in the tables. Same with S2; also it’s not clear why some of the same data which has been included in Raw Data files has been included under S2 as well.

-Also in S2, please make the legends self-explanatory as to what data has been included in the table. Also in the caption “Raw data for the different anther development stages”, what are the development stages you are referring to here, also it is not very explicitly clear why the study starts focusing on anthers.

The introduction needs revisions to make it easier and clearer for the reader to understand. Complex sentences throughout the section need to be broken down into simpler sentences for easier understanding.

“unpublished transcriptome data from the laboratory”; Need to explicitly mention what was the objective to obtain this data so that it is clear what is the nature of the data.

“The physical properties of WRKY members, such as length, molecular
94 weight (MW), and isoelectric point (pI), were predicted using the online ExPasy program” List all the features found as separate Supplementary Table and cite it here.

The way sub-sections are arranged in the Methods section makes it unclear. Please compile the first two sections under “Computational analysis …” and then the next section as “Experimental validation..”
Need more content under the conclusion. The font size and style are not uniform throughout the manuscript.

References need to be adequately formatted.
Figure 3: Legends should be self-explanatory. Include how many species are being denoted in the tree.
Combine Figures 7-10 as subparts of one figure.

Experimental design

Experimental design is good to start with, but it would be more interesting to include more analyses that are more insightful about the WRKY family.In the current state, the authors have focussed on carrying out phylogenetic analysis to create trees, and generate protein interaction networks using bioinformatics software, but haven't made any strong conclusions from the study. It will be more helpful if they can make sure which WRKY subspecies are more closely related and how that affected their function etc.I would be interested if they can add more discussion pertaining to how the phylogenetic trees,network analysis they have created will be helpful in future studies.

Validity of the findings

They have added raw data and other files to support their findings.I have a few comments regarding how they are sharing the data. Please make sure the legends of the files are well-explained so that is easier to catch up and reproduce results if readers attempt themselves.

Additional comments

I have included more comments as pdf attached.

Annotated reviews are not available for download in order to protect the identity of reviewers who chose to remain anonymous.

---

## Round 0.2 · Minor Revisions

Please address remaining concerns of the reviewer

Reviewer 3 ·

Basic reporting

The article now has incorporated many changes suggested which makes it a much better version. Although, there are a few issues still remaining which I have included as comments in the pdf attached.

Experimental design

The experimental design is well-defined and meaningful.

Validity of the findings

I think the authors have well-explained and supported their observations and findings.

Additional comments

Attached as pdf.

Annotated reviews are not available for download in order to protect the identity of reviewers who chose to remain anonymous.

---

## Round 0.3 · accepted · Accept

All remaining issues are addressed and therefore revised manuscript is acceptable now.